# Association of Ventilatory Disorders with Respiratory Symptoms, Physical Activity, and Quality of Life in Subjects with Prior Tuberculosis: A National Database Study in Korea

**DOI:** 10.3390/jpm11070678

**Published:** 2021-07-19

**Authors:** Bumhee Yang, Hayoung Choi, Sun Hye Shin, Youlim Kim, Ji-Yong Moon, Hye Yun Park, Hyun Lee

**Affiliations:** 1Division of Pulmonary and Critical Care Medicine, Department of Internal Medicine, Chungbuk National University Hospital, Chungbuk National University College of Medicine, Cheongju 28644, Korea; ybhworld0415@gmail.com; 2Division of Pulmonary, Allergy, and Critical Care Medicine, Department of Internal Medicine, Hallym University Kangnam Sacred Heart Hospital, Hallym University College of Medicine, Seoul 07441, Korea; hychoimd@gmail.com; 3Division of Pulmonary and Critical Care Medicine, Department of Medicine, Samsung Medical Center, Sungkyunkwan University School of Medicine, Seoul 06351, Korea; freshsunhye@gmail.com; 4Division of Pulmonary, Allergy, and Critical Care Medicine, Department of Internal Medicine, Hallym University Chuncheon Sacred Heart Hospital, Chuncheon 24252, Korea; weilin810707@gmail.com; 5Lung Research Institute, Hallym University College of Medicine, Chuncheon 24252, Korea; 6Division of Pulmonary Medicine and Allergy, Department of Internal Medicine, Hanyang University College of Medicine, Seoul 04763, Korea; fattymoon@gmail.com

**Keywords:** tuberculosis, pulmonary function, respiratory symptoms, quality of life

## Abstract

Tuberculosis (TB) survivors experience post-TB lung damage and ventilatory function disorders. However, the proportions of obstructive and restrictive ventilatory disorders as well as normal ventilation among subjects with prior TB are unknown. In addition, the impacts of ventilatory disorder and its severity on respiratory symptoms, physical activity limitations, and the quality of life in subjects with prior TB remain unclear. Subjects who participated in the Korean National Health and Nutritional Examination Survey 2007–2016 were enrolled in this study. We evaluated the impact of each ventilatory disorder and its severity on respiratory symptoms, physical activity limitations, and quality of life (measured by the EuroQoL five dimensions questionnaire [EQ-5D] index values) in subjects with prior TB. Among 1466 subjects with prior TB, 29% and 16% had obstructive ventilatory disorders and restrictive ventilatory disorders, respectively. Mild and moderate obstructive ventilatory disorders were not associated with respiratory symptoms, physical activity limitations, or EQ-5D index value compared with normal ventilation; however, severe obstructive ventilatory disorders were associated with more respiratory symptoms (adjusted odds ratio [aOR] = 13.62, 95% confidence interval [CI] = 4.64–39.99), more physical activity limitation (aOR = 218.58, 95% CI = 26.82–1781.12), and decreased EQ-5D index (adjusted coefficient = −0.06, 95% CI = (−0.12–−0.10) compared with normal ventilation. Mild restrictive ventilatory disorders were associated with more respiratory symptoms (aOR = 2.10, 95% CI = 1.07–4.14) compared with normal ventilation, while moderate (aOR = 5.71, 95% CI = 1.14–28.62) and severe restrictive ventilatory disorders (aOR = 9.17, 95% CI = 1.02–82.22) were associated with physical activity limitation compared with normal ventilation. In conclusion, among subjects with prior TB, 29% and 16% developed obstructive and restrictive ventilatory disorders, respectively. Severe obstructive ventilatory disorder was associated with more respiratory symptoms, more physical activity limitation, and poorer quality of life, while severe restrictive ventilatory disorder was associated with more physical activity limitations.

## 1. Introduction

Tuberculosis (TB) is the leading cause of death from a single epidemic disease worldwide [1]. Since recognition of TB as a deadly communicable disease, the mortality rate has decreased, and 85% of patients treated for their first episode of TB survive with treatment success (cure or completion) [1,2]. Despite the improved survival rate, approximately 50% of TB survivors experience post-TB lung damage and ventilatory disorders [3,4,5]. Depending on the severity of post-TB lung damage, they show various clinical courses, including no respiratory symptoms, dyspnea, or impaired quality of life (QoL) [6,7].

The ventilatory disorders of subjects with prior TB comprise obstructive and restrictive patterns [8]. Obstructive ventilatory disorder, which appears as airflow limitation on pulmonary function tests, is the most well-known form of lung damage after TB treatment [5,8]. Post-TB lung damage, including cavity, bronchiectasis, or distorted airway, can cause obstructive ventilatory disorder, which can lead to dyspnea, chronic obstructive pulmonary disease, and reduced exercise capacity [9,10]. Although obstructive disorder has been recognized as post-TB lung damage, restrictive ventilatory disorder is also found in subjects with prior TB and can lead to dyspnea or chest pain [11]. The restrictive pattern has been suggested to be a consequence of excessive fibrosis, fibrotic bands, or bronchovascular distortion in the process of post-TB lung repair [12]. The proportions of obstructive and restrictive ventilatory disorders as well as normal ventilation among subjects with prior TB remain unclear. Additionally, the impacts of ventilatory defects and severity of defects on respiratory symptoms, physical limitations, and QoL in subjects with prior TB have not been investigated.

This study sought to determine the composition of ventilatory defects among subjects with prior TB using a nationally representative database in South Korea. Furthermore, this study investigated the impact of each ventilatory disorder and its severity on respiratory symptoms, physical activity limitations, and QoL among subjects with prior TB.

## 2. Material and Methods

### 2.1. Study Population

We used the data from the 2007–2016 Korea National Health and Nutrition Examination Survey (NHANES), a nationally representative health survey collected by the Korean Ministry of Health and Welfare. Health-related questionnaires, health examinations, and spirometry results were used in this study. Previous pulmonary TB was defined based on formal reading of a chest X-ray or a history of physician diagnosed pulmonary TB. We classified subjects with prior TB into three groups according to spirometric pattern (Figure 1). The study protocol was approved by the Institutional Review Board of Chungbuk National University Hospital (application no. 2021-01-041).

### 2.2. Measurements

Data on age, sex, body mass index (BMI), smoking history, physical activity limitations, occupation, EuroQoL five dimensions questionnaire (EQ-5D) index value, and spirometric results were obtained from the Korea NHANES database. Respiratory symptoms, including cough, sputum, and dyspnea, were measured qualitatively (presence or absence). To assess physical activity limitations, we used a questionnaire, “Do you experience physical activity limitations due to respiratory disease?”, and this was also measured qualitatively (yes or no).

The EQ-5D enables the respondent to classify his or her health according to five dimensions. These dimensions define health in terms of mobility, self-care, usual activity, pain/discomfort, and anxiety/depression. Each dimension is divided into three levels, i.e., no problem/some or moderate problems/extreme problems. The information derived from the EQ-5D self-classifier can be converted into a single summary index: the EQ-5D index [13]. The EQ-5D index ranges between 0 (worst imaginable health state) and 1 (best imaginable health state).

Spirometry was performed according to the recommendations of the American Thoracic Society and European Respiratory Society [14]. Since post-bronchodilator spirometry was not available in Korea NHANES database, pre-bronchodilator spirometry results were used in our study. Absolute values of forced expiratory volume in 1 s (FEV_1_) and forced vital capacity (FVC) were obtained, and the percentage of predicted values (% predicted) for FEV_1_ and FVC were calculated using the reference equation obtained on analysis of a representative Korean sample [15]. Comorbidities of asthma, diabetes mellitus, hypertension, dyslipidemia, cardiovascular disease, osteoporosis, osteoarthritis or rheumatoid arthritis, and depression were self-reported based on previous physician diagnosis [16].

### 2.3. Definitions of Ventilatory Disorder

Normal ventilation was defined as pre-bronchodilator FEV_1_/FVC ≥ 0.70 and FVC ≥ 80% predicted. Obstructive ventilatory disorder was defined as pre-bronchodilator FEV_1_/FVC < 0.70 [17]. For cases with obstructive ventilatory disorder, FEV_1_ ≥ 80% predicted, FEV_1_ of 50–79% predicted, and FEV_1_ < 50% predicted were classified as mild, moderate, and severe, respectively. Restrictive ventilatory disorder was defined as FEV_1_/FVC ≥ 0.7 and FVC < 80% predicted. For cases with restrictive ventilatory disorder, FVC ≥ 70% predicted, FVC of 60–69% predicted, and FVC < 60% predicted were classified as mild, moderate, and severe, respectively [18,19].

### 2.4. Outcomes

We compared respiratory symptoms, physical activity limitations due to respiratory diseases (hereafter physical activity limitations), and QoL (measured using the EQ-5D index) between subjects with prior TB with different ventilatory disorders. We also analyzed the impacts of ventilatory disorder severity on respiratory symptoms, physical activity limitations, and QoL in subjects with prior TB.

### 2.5. Statistical Analysis

All analysis was performed using survey commands in STATA 15.1 version (StataCorp LP, College Station, TX, USA) to account for the complex sampling design and survey weights. For each variable, we calculated prevalence and 95% confidence interval (CI) by group.

The associations between ventilatory disorders and respiratory symptoms (cough, sputum, or dyspnea) and physical activity limitations were analyzed using logistic regression analysis: Multivariable analysis was adjusted for age (categorized as ≥65 years or not), sex, BMI, smoking amount (pack-years), education level (categorized as high school or less vs. college or above), and family income (categorized as low vs. high). Linear regression analysis was performed to assess the association between ventilatory disorders and EQ-5D index scores: Multivariable analysis was adjusted for covariates as mentioned above.

A two-sided *p* value < 0.05 was considered significant. To account for multiple comparisons, post hoc Bonferroni correction was applied (normal ventilation vs. obstructive ventilatory disorder, normal ventilation vs. restrictive ventilatory disorder, and obstructive ventilatory disorder vs. restrictive ventilatory disorder), in which a *p* value of 0.05 corresponds to 0.017 (0.05/3).

## 3. Results

### 3.1. Baseline Characteristics

As shown in Table 1, the post-TB patient group included 1466 patients (54.9%) with normal ventilation, 783 patients (29.3%) with obstructive ventilatory disorders, and 420 patients (15.8%) with restrictive ventilatory disorders. The mean ages of subjects with normal ventilation, obstructive ventilatory disorders, and restrictive ventilatory disorders were 53.4, 64.4, and 59.6 years, respectively (*p* < 0.001). The proportion of males was highest in subjects with obstructive ventilatory disorders, followed by those with restrictive ventilatory disorders and those with normal ventilation (76.2%, 52.0%, and 49.5%, respectively, *p* < 0.001). The subjects with obstructive ventilatory disorders had a lower BMI than those with normal ventilation or restrictive ventilatory disorders (22.8 kg/m^2^, 23.7 kg/m^2^, and 23.8 kg/m^2^, respectively, *p* < 0.001). The subjects with obstructive ventilatory disorders had higher rates of current or ex-smoking than those with normal ventilation or restrictive ventilatory disorders (71.8%, 44.5%, and 39.4%, respectively, *p* < 0.001). The proportion of subjects with low family income was highest among obstructive ventilatory disorders, followed by restrictive ventilatory disorders, and normal ventilation (68.3%, 55.9%, and 43.0%, respectively, *p* < 0.001). The subjects with obstructive ventilatory disorders had the highest prevalence of asthma (10.9%), diabetes mellitus (19.1%), and hypertension (50.8%) among subjects with prior TB, while subjects with restrictive ventilatory disorders had the highest prevalence of dyslipidemia (48.4%) and osteoporosis (11.9%). Regarding spirometric results, FVC (L) and FVC % predicted were lowest in subjects with restrictive ventilatory disorders, while FEV_1_ (L), FEV_1_ % predicted, and FEV_1_/FVC were lowest in those with obstructive ventilatory disorders among subjects with prior TB (*p* < 0.001 for all).

### 3.2. Comparison of Symptoms, Physical Activity, and Quality of Life

As shown in Table 2, respiratory symptoms including sputum (18.2%, 15.1%, and 9.5%, respectively, *p* = 0.004) and dyspnea (3.8%, 2.4%, and 0.9%, respectively, *p* < 0.020) and physical activity limitations (27.7%, 13.0%, and 5.1%, respectively, *p* < 0.001) were most frequently observed in subjects with obstructive ventilatory disorders, followed by those with restrictive ventilatory disorders, and those with normal ventilation. Cough was observed most frequently in subjects with obstructive ventilatory disorders followed by those with normal ventilation, and those with restrictive ventilatory disorders (11.6%, 5.4%, and 5.1%, respectively, *p* < 0.001).

The EQ-5D index values, denoting QoL, were lower among subjects with obstructive ventilatory disorders and restrictive ventilatory disorders than in those with normal ventilation (0.91, 0.91, and 0.94, respectively, *p* = 0.002). Regarding the individual EQ-5D component arm, the subjects with obstructive ventilatory disorders had the highest rates of mobility difficulty (24.1%, *p* < 0.001) and self-care limitations (7.7%, *p* = 0.005) among subjects with prior TB; however, those with restrictive ventilatory disorders had the highest rates of difficulty with usual activities (19.1%, *p* < 0.001) and pain/discomfort (36.3%, *p* = 0.015) among subjects with prior TB (Table 2).

### 3.3. The Impact of Obstructive Ventilatory Disorder and Its Severity on Respiratory Symptoms, Physical Activity Limitations, and EQ-5D Index in Subjects with Prior TB

As shown in Appendix A, the subjects with obstructive ventilatory disorders were 1.63 (95% CI = 1.05–2.82) times more likely to have any respiratory symptoms compared to those with normal ventilation in the fully adjusted model; additionally, obstructive ventilatory disorders were closely associated with sputum (aOR = 1.85, 95% CI = 1.03–3.37) and dyspnea (aOR = 4.19, 95% CI = 1.03–17.14). Likewise, the subjects with obstructive ventilatory disorder had more physical activity limitations (aOR = 6.59, 95% CI = 1.98–21.93) compared to those with normal ventilation. Although subjects with obstructive ventilatory disorders were more likely to have a lower EQ-5D index compared to those with normal ventilation in the unadjusted model (coefficient = −0.02, 95% CI −0.04–−0.07), this was not significant in the adjusted model. 

In the analyses according to obstructive ventilatory disorder severity, mild and moderate obstructive ventilatory disorders were not associated with more respiratory symptoms, more physical activity limitations, or lower EQ-5D index values in the unadjusted and adjusted models. However, severe obstructive ventilatory disorder was significantly associated with more respiratory symptoms (aOR = 13.62, 95% CI = 4.64–39.99), more physical activity limitations (aOR = 218.58, 95% CI = 26.82–1781.12), and lower EQ-5D index values (adjusted coefficient = −0.06, 95% CI = −0.12–−0.01) (Table 3).

### 3.4. The Impact of Restrictive Ventilatory Disorder and Its Severity on Respiratory Symptoms, Physical Activity Limitations, and EQ-5D Index Value in Subjects with Prior TB

Restrictive ventilatory disorder was not significantly associated with increased respiratory symptoms (cough, sputum, or dyspnea), physical activity limitations, and EQ-5D index values compared with normal ventilation in adjusted models (Appendix A).

In the analyses according to restrictive ventilatory disorder severity, the subjects with mild restrictive ventilatory disorders were more likely to have respiratory symptoms (aOR = 2.10, 95% CI = 1.07–4.14) compared to those with normal ventilation, specifically sputum (aOR = 2.80, 95% CI = 1.34–5.83). In comparison, moderate (aOR = 5.71, 95% CI = 1.14–28.62) and severe (aOR = 9.17, 95% CI = 1.02–82.22) restrictive ventilatory disorders were associated with more physical activity limitations compared to those with normal ventilation (Table 4).

## 4. Discussion

To the best of our knowledge, this is the first study evaluating respiratory symptoms, physical activity limitations, and QoL according to ventilatory disorder type and severity in subjects with prior TB. Among subjects with prior TB, approximately 29% and 16% developed obstructive and restrictive ventilatory disorders, respectively. Severe obstructive ventilatory disorders were associated with more respiratory symptoms, more physical activity limitations, and poorer quality of life. Severe restrictive ventilatory disorder was associated with more physical activity limitations.

TB survivors frequently experience structural and functional lung sequelae that vary in severity [7]. For example, approximately 24–35% of TB survivors experience obstructive ventilatory disorders [20,21,22]. In agreement with previous reports, 29% of the subjects with prior TB in this study had obstructive ventilatory disorders. Thus, development of obstructive ventilatory disorders can cause a concerning health-related burden in TB survivors. Subjects with prior TB also can experience restrictive ventilatory disorders. Post-TB survivors often show a fibrotic pattern on chest imaging due to the sequelae of pulmonary TB including destruction of lung parenchyma [23]. Restrictive ventilatory disorders occur in subjects with prior TB due to volume loss, lung scarring with reduction of pulmonary compliance, and an increase in elastic retraction pressure [24,25]. In contrast to the literature on obstructive ventilatory disorders, only a few studies have described restrictive ventilatory disorders, and the prevalence was reported to be 31–42% among TB survivors [26,27]. The small number of patients in these studies (*n* = 107 and *n* = 33) limited the study findings [26,27]. Thus, our study had the advantage of confirming these findings with a larger number of subjects using a nationwide database.

As shown in previous studies [20,28], clinical factors associated with poorer QoL (e.g., old age, male sex, smoking history, lower BMI, and lower education level) were more common in subjects with prior TB with obstructive ventilatory disorders than in those with normal ventilation. Approximately 72% of subjects with prior TB with obstructive ventilatory disorders in this study were current or ex-smokers. In line with this finding, smoking is a well-established factor associated with obstructive ventilatory disorders in subjects with prior TB [20]. However, little is known as to whether development of obstructive ventilatory disorders associated with higher symptomatic burdens in post-TB survivors, as most previous studies focused on the presence of obstructive ventilatory disorders and their severity after TB treatment [5,11,29]. From this perspective, our study has confirmed that the development of severe obstructive ventilatory disorders is associated with more respiratory symptoms, more physical activity limitations, and poorer QoL compared to patients with normal ventilation; however, these findings were not significant in patients with mild-to-moderate obstructive ventilatory disorders. These results indicate that regular health check-ups with pulmonary function measurement is necessary after completion of TB treatment to detect obstructive ventilatory disorders early and provide appropriate treatment to prevent further lung function impairment. Recent studies also support this suggestion in showing clinical improvement after bronchodilator treatment in TB-destroyed lung patients with obstructive ventilatory disorder [30,31].

Despite the prevalence of restrictive ventilatory disorders after TB, to the best of our knowledge, no studies have evaluated the association between restrictive ventilatory disorder and respiratory symptoms, physical activity limitations, and QoL in TB survivors. Our study revealed that respiratory symptoms and QoL were not significantly affected by restrictive ventilatory disorders in subjects with prior TB, while physical activity limitations were significant in subjects with prior TB with moderate-to-severe restrictive ventilatory disorders. Restrictive ventilatory disorders might be an underappreciated cause of functional impairments and respiratory symptoms [32,33]. One study showed that 35.4% of subjects with restrictive ventilatory disorders reported at least one chronic respiratory symptom [33]. One reason why our study results contrast with previous findings might be that most subjects with restrictive ventilatory disorders in our study had a mild degree of restrictive abnormality; thus, the number of subjects with moderate-to-severe restrictive ventilator disorders was relatively small. Accumulating evidence has shown that restrictive ventilatory disorders are related to physical activity limitations, which is in line with our study results [34,35]. The significant association of moderate-to-severe restrictive ventilatory disorders with physical activity limitations, but not with respiratory symptoms, suggests that restrictive ventilatory disorders influence physical activity limitations through mechanisms that are at least partly independent of respiratory symptoms. Decreased lung or chest wall compliance and increased elastic work of breathing might be one mechanism underlying physical activity limitations in patients with advanced restrictive ventilatory disorders [36].

This study has several limitations. First, this study was performed among a representative sample of Koreans. Thus, our data might not be generalizable to other ethnic groups or populations. Second, obstructive ventilatory disorders were defined by pre-bronchodilator spirometric results. This might lead to overestimation of the prevalence of obstructive ventilatory disorders; however, our estimates were similar to previous studies [20]. Third, the relatively small number of subjects with prior TB with moderate-to-severe restrictive ventilatory disorders might lead to statistical nonsignificance when analyzing the impact of ventilatory disorder severity on respiratory symptoms or QoL. Fourth, because this retrospective study used the Korea NHANES database that had been established before we designed this study, power calculations for sample size could not be performed during the design phase of the study. Fifth, there is a possibility that some subjects already had ventilatory disorders before *Mycobacterium tuberculosis* infection; however, in the Korea NHANES database, spirometry results before TB were not available. Sixth, as the Korea NHANES database does not provide some important clinical information that can affect respiratory symptoms and quality of life, such as medication use or exacerbation history, we could not adjust for these factors.

## 5. Conclusions

Among TB survivors, 29% had obstructive ventilatory disorders and 16% had restrictive ventilatory disorders. Severe obstructive ventilatory disorders were associated with an increased health-related burden, including more respiratory symptoms, more physical activity limitations, and poorer QoL, while severe restrictive ventilatory disorders were associated with more physical activity limitations. Further research is needed to establish strategies for early diagnosis and adequate treatment of ventilatory disorders in TB survivors.

## Figures and Tables

**Figure 1 jpm-11-00678-f001:**
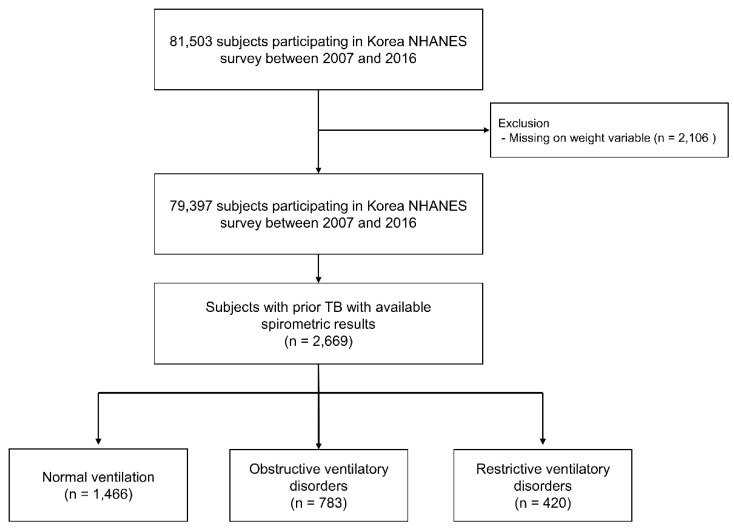
Flow chart of patient selection. TB, Tuberculosis; NHANES, National Health and Nutrition Examination Survey.

**Table 1 jpm-11-00678-t001:** Baseline characteristics of subjects with prior TB by spirometric pattern.

	Total (*n* = 2669)	Normal Ventilation (*n* = 1466)	Obstructive Ventilatory Disorder (*n* = 783)	Restrictive Ventilatory Disorder (*n* = 420)	*p* Value
Age, years	57.5 (56.7–58.3)	53.4 (52.5–65.5)	64.4 (63.2–65.5) ^a^	59.6 (57.7–61.4) ^b,c^	<0.001
Male sex	57.7 (55.2–60.2)	49.5 (45.9–53.1)	76.2 (72.1–79.8) ^a^	52.0 (45.2–58.8) ^c^	<0.001
BMI, kg/m^2^	23.4 (23.3–23.6)	23.7 (23.5–23.9)	22.8 (22.5–23.1) ^a^	23.8 (23.4–24.3) ^c^	<0.001
Smoking history					<0.001
Never-smoker	48.2 (45.6–50.8)	55.5 (51.8–59.2)	28.2 (24.1–32.6) ^a^	60.6 (53.5–67.2) ^b,c^	
Current- or ex-smoker	51.8 (49.2–54.4)	44.5 (40.8–59.2)	71.8 (67.4–75.9) ^a^	39.4 (32.8–46.5) ^b,c^	
Smoking amount, pack-years	7.3 (6.3–8.2)	5.5 (4.4–6.6)	13.3 (10.9–15.5) ^a^	5.4 (3.4–7.4) ^b,c^	
Family income					<0.001
Low	52.1 (49.2–55.0)	43.0 (39.2–46.8)	68.3 (63.4–72.8) ^a^	55.9 (49.0–62.6) ^b^	
High	47.9 (45.0–50.8)	57.0 (53.2–60.8)	31.7 (27.2–36.6) ^a^	44.1 (37.4–51.0) ^b^	
Education					<0.001
High school or less	79.1 (76.7–81.3)	74.9 (71.6–78.0)	86.5 (82.5–89.7) ^a^	80.4 (73.9–89.7)	
More than high school	20.9 (18.7–23.3)	25.1 (22.0–28.4)	13.5 (10.1–17.5) ^a^	19.6 (14.4–26.1)	
Comorbidities					
Asthma	4.9 (3.9–6.1)	2.3 (1.4–3.6)	10.9 (8.3–14.1) ^a^	2.8 (1.4–5.3) ^b,c^	<0.001
Diabetes mellitus	12.4 (10.7–14.3)	8.3 (6.5–10.5)	19.1 (15.5–23.4) ^a^	15.0 (10.9–20.2) ^b^	<0.001
Hypertension	41.4 (38.7–44.1)	35.0 (31.7–38.6)	50.8 (45.7–55.8) ^a^	47.2 (40.4–54.1) ^b^	<0.001
Dyslipidemia	44.1 (41.4–47.0)	42.9 (39.2–46.7)	44.5 (39.5–49.7)	48.4 (41.5–55.3)	0.413
Cardiovascular disease	5.1 (4.0–6.5)	4.5 (3.3–6.2)	5.3 (3.5–7.9)	7.3 (4.1–12.6)	0.287
Osteoporosis	8.2 (6.5–10.3)	7.4 (5.5–10.3)	7.7 (4.6–12.5)	11.9 (7.3–19.0)	0.297
Arthritis	15.1 (13.4–16.9)	15.4 (12.4–18.0)	12.9 (10.3–16.0)	18.3 (13.9–23.8)	0.140
Depression	4.5 (3.5–5.6)	4.4 (3.2–6.1)	4.6 (2.9–7.1)	4.4 (2.4–7.7)	0.993
Spirometry					
FVC, L	3.5 (3.4–3.5)	3.7 (3.6–3.8)	3.4 (3.3–3.5) ^a^	2.7 (2.6–2.8) ^b,c^	<0.001
FVC, % predicted	88.7 (88.0–89.4)	94.4 (93.6–95.1)	85.5 (84.1–86.9) ^a^	72.6 (71.5–73.7) ^b,c^	<0.001
FEV_1_, L	2.6 (2.5–2.6)	2.9 (2.8–3.0)	2.0 (2.0–2.1) ^a^	2.1 (2.0–2.2) ^b,c^	<0.001
FEV_1_, % predicted	84.7 (83.8–85.5)	94.4 (93.6–95.1)	70.8 (69.3–72.4) ^a^	74.3 (73.1–75.6) ^b,c^	<0.001
FEV_1_/FVC	73.3 (72.8–74.0)	78.9 (78.6–79.3)	60.6 (59.7–61.4) ^a^	78.0 (77.2–78.7)^b,c^	<0.001

Data are presented as weighted mean (95% confidence interval) or weighted percentage (95% confidence interval). *p* values are comparisons of three groups. TB, tuberculosis; BMI, body mass index; FVC, forced vital capacity; FEV_1_, forced expiratory volume in 1 s. The results of Bonferroni correction with three comparisons are provided as superscripts in Table (a *p* value of 0.05 corresponds to 0.017 [0.05/3]). ^a^ Indicates statistical significance for the comparison of normal ventilation and obstructive ventilatory disorder. ^b^ Indicates statistical significance for the comparison of normal ventilation and restrictive ventilatory disorder. ^c^ Indicates statistical significance for the comparison of obstructive ventilatory disorder and restrictive ventilatory disorder.

**Table 2 jpm-11-00678-t002:** Comparison of symptoms, physical activity, and quality of life according to spirometric pattern.

	Total (*n* = 2669)	Normal Ventilation (*n* = 1466)	Obstructive Ventilatory Disorder (*n* = 783)	Restrictive Ventilatory Disorder (*n* = 420)	*p* Value
Any respiratory symptoms	16.4 (14.0–19.3)	12.4 (9.5–16.1)	22.9 (18.1–28.5) ^a^	18.6 (12.6–26.6)	0.001
Cough	7.3 (5.7–9.2)	5.4 (3.5–8.3)	11.6 (8.5–15.8) ^a^	5.1 (2.8–9.3)	0.004
Sputum	13.0 (10.8–15.5)	9.5 (7.0–12.8)	18.2 (13.9–23.5) ^a^	15.1 (9.5–23.2)	0.004
Dyspnea	2.0 (1.3–3.1)	0.9 (0.3–2.5)	3.8 (2.1–6.7)	2.4 (0.9–5.9)	0.020
Physical activity limitations	15.6 (11.2–21.2)	5.1 (2.3–10.9)	27.7 (19.3–38.0) ^a^	13.0 (6.2–28.4)	<0.001
EQ-5D component					
Mobility	18.3 (16.4–20.3)	14.0 (11.9–16.4)	24.1 (20.4–28.1) ^a^	23.5 (18.1–29.8) ^b^	<0.001
Self-care	5.2 (4.3–6.6)	3.6 (2.6–5.0)	7.7 (5.5–10.7) ^a^	7.3 (4.3–12.4)	0.005
Usual activity	12.8 (11.1–14.6)	9.4 (7.7–11.5)	16.3 (13.1–20.1) ^a^	19.1 (14.2–25.2) ^b^	<0.001
Pain/discomfort	29.0 (26.6–31.5)	30.7 (26.5–35.2)	30.1 (26.5–35.2)	36.3 (29.8–43.4)	0.015
Anxiety/depression	13.6 (11.9–15.5)	13.5 (11.3–16.1)	13.5 (10.6–17.1)	13.7 (9.5–17.5)	0.999
EQ-5D index	0.93 (0.92–0.93)	0.94 (0.93–0.94)	0.91 (0.90–0.93) ^a^	0.91 (0.89–0.93) ^b,c^	0.002

Data are presented as weighted mean (95% confidence interval) or weighted percentage (95% confidence interval). *p* values are comparisons of three groups. EQ-5D, EuroQoL five dimensions. The results of Bonferroni correction with three comparisons are provided as superscripts in Table (a *p* value of 0.05 corresponds to 0.017 [0.05/3]). ^a^ Indicates statistical significance for the comparison of normal ventilation and obstructive ventilatory disorder. ^b^ Indicates statistical significance for the comparison of normal ventilation and restrictive ventilatory disorder. ^c^ Indicates statistical significance for the comparison of obstructive ventilatory disorder and restrictive ventilatory disorder.

**Table 3 jpm-11-00678-t003:** The impact of obstructive ventilatory disorder severity on respiratory symptoms, physical activity limitations, and EQ-5D index value in subjects with prior TB.

	Model	Normal (*n* = 1466)	Obstructive Ventilatory Disorder		
			Mild (*n* = 256)	Moderate (*n* = 432)	Severe (*n* = 95)
Respiratory symptoms	Crude model	Reference	1.52 (0.83, 2.81)	1.67 (1.03, 2.71)	8.38 (4.16, 16.88)
	Adjusted model *	Reference	1.09 (0.51, 2.33)	1.05 (0.53, 2.08)	13.62 (4.64, 39.99)
Cough	Crude model	Reference	2.45 (1.08, 5.56)	1.74 (0.87, 3.46)	4.99 (2.04, 12.28)
	Adjusted model *	Reference	1.81 (0.69, 4.76)	1.13 (0.47, 2.75)	3.87 (1.14, 13.21)
Sputum	Crude model	Reference	1.71 (0.91, 3.22)	1.69 (1.01, 2.84)	6.84 (3.24, 14.44)
	Adjusted model *	Reference	1.34 (0.59, 3.05)	1.18 (0.56, 2.51)	11.39 (3.17, 40.94)
Dyspnea	Crude model	Reference	1.95 (0.43, 8.88)	2.12 (0.56, 7.98)	21.58 (5.51, 84.46)
	Adjusted model *	Reference	1.41 (0.30, 6.72)	3.56 (0.68, 18.67)	21.42 (3.50, 13.13)
Physical activity limitations	Crude model	Reference	3.55 (0.81, 15.46)	3.60 (1.25, 10.38)	38.35 (10.80, 136.15)
	Adjusted model *	Reference	7.83 (0.99, 61.27)	5.59 (0.93, 33.51)	218.58 (26.82, 1781.12)
EQ-5D index	Crude model	Reference	−0.01 (−0.26, 0.13)	−0.02 (−0.04, −0.001)	−0.07 (−0.11, −0.02)
	Adjusted model *	Reference	0.01 (−0.01, 0.04)	−0.003 (−0.03, 0.02)	−0.06 (−0.12, −0.01)

Data are presented as a ratio (95% confidence interval) or a difference estimate (95% confidence interval). * Age, sex, body mass index, smoking amount (pack-years), education (categorized as >high school or ≤high school), and family income (categorized as low or high) were adjusted. EQ-5D, EuroQoL five dimensions; TB, tuberculosis.

**Table 4 jpm-11-00678-t004:** The impact of restrictive ventilatory disorder severity on respiratory symptoms, physical activity limitations, and EQ-5D index value in subjects with prior TB.

	Model	Normal (*n* = 1466)	Restrictive Ventilatory Disorder		
			Mild (*n* = 306)	Moderate (*n* = 77)	Severe (*n* = 37)
Respiratory symptoms	Crude model	Reference	1.95 (1.09, 3.49)	0.95 (0.30, 3.06)	0.59 (0.11, 3.24)
	Adjusted model *	Reference	2.10 (1.07, 4.14)	1.10 (0.28, 4.37)	0.87 (0.14, 5.55)
Cough	Crude model	Reference	0.96 (0.39, 2.38)	0.77 (0.16, 3.80)	1.11 (0.14, 8.98)
	Adjusted model *	Reference	0.57 (0.19, 1.66)	0.85 (0.14, 4.82)	1.34 (0.14, 12.55)
Sputum	Crude model	Reference	2.23 (1.16, 4.26)	0.42 (0.09, 2.00)	0.61 (0.08, 4.77)
	Adjusted model *	Reference	2.80 (1.34, 5.83)	0.46 (0.08, 2.72)	0.84 (0.08, 8.70)
Dyspnea	Crude model	Reference	1.20 (0.21, 6.82)	8.56 (1.43, 51.34)	1.94 (0.20, 18.40)
	Adjusted model *	Reference	0.98 (0.12, 7.94)	18.20 (3.05, 108.58)	3.04 (0.17, 53.88)
Physical activity limitation	Crude model	Reference	2.29 (0.46, 11.38)	3.58 (0.71, 18.02)	5.34 (0.85, 33.71)
	Adjusted model *	Reference	1.92 (0.20, 18.54)	5.71 (1.14, 28.62)	9.17 (1.02, 82.22)
EQ-5D index	Crude model	Reference	−0.01 (−0.03, 0.01)	−0.09 (−0.17, −0.01)	−0.05(−0.09, −0.001)
	Adjusted model *	Reference	−0.01(−0.21, 0.02)	−0.08 (−0.18, 0.02)	−0.021 (−0.08, 0.04)

Data are presented as a ratio (95% confidence interval) or a difference estimate (95% confidence interval). * Adjusted for age, sex, body mass index, smoking amount (pack-years), education (categorized as >high school or ≤high school), and family income (categorized as low or high). EQ-5D, EuroQoL five dimensions; TB, tuberculosis.

## Data Availability

The data presented in this study are available on request from the corresponding author.

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
