# Peer review of "Association of Ventilatory Disorders with Respiratory Symptoms, Physical Activity, and Quality of Life in Subjects with Prior Tuberculosis: A National Database Study in Korea"

_jpm, 2021, doi:10.3390/jpm11070678_

Round 1
Reviewer 1 Report
the study is database, population based and hence very interesting
also the topic on TB pulmonary sequelae and their clinical and functional impact very interesting
Major flaws
- Methodology: it is not clear if you only measured respiratory symptoms qualitatively ( ie present/absent) or you applied a severity scale. if the latter is the case please describe. if the former please specify it in the methodology. Physical activity: how did you measure the degree of its impairment? please describe. It is not clear how the scores for the Euroqol 5D were calculated.
- Results regressions in tables 3 and 4 are misinterpreted. for example for obstruction, people with cough were 1.69 more likely to have cough irrespective of the severity of this dysfunction. the same for the other variables introduced in your model. the same for restriction. The titles of the tables 3 for are not reflecting the content of them. Table 2 does not mention for which comparison set(ie normal and obstructive? normal and restrictive? obstructive and restrictive? all three? is the p value mentioned
Author Response
## Response to Reviewer 1’s comments
Specific comments
Comment 1 (C1). Methodology: it is not clear if you only measured respiratory symptoms qualitatively ( ie present/absent) or you applied a severity scale. if the latter is the case please describe. if the former please specify it in the methodology.
Response 1 (R1). Thank you for your comments. Respiratory symptoms, including cough, sputum and dyspnea, were only measured qualitatively: presence or absence. We have clarified this in the Methods section of the revised manuscript (page 3, 1st paragraph).
“Respiratory symptoms, including cough, sputum and dyspnea, were measured qualita-tively: presence or absence.”
C2. Physical activity: how did you measure the degree of its impairment? please describe.
R2. The Korea NHANES has a questionnaire, “Do you experience physical activity limitations due to respiratory diseases?”, and this was also only measured qualitatively: yes or no. We have added this information in in the Methods section of the revised manuscript (page 3, 1st paragraph).
“To assess physical activity limitations, we used a questionnaire, “Do you experience phys-ical activity limitations due to respiratory disease?”, and this was also measured qualita-tively: yes or no.”
C3. It is not clear how the scores for the Euroqol 5D were calculated.
R3. Thank you for pointing this out, which we did not fully acknowledge in our original manuscript. The EQ-5D enables the respondent to classify his/her health according to five dimensions. These dimensions define health in terms of mobility, self-care, usual activity, pain/discomfort, and anxiety/depression. Each dimension is divided into three level, i.e., no problem/some or moderate problems/extreme problems. The information derived from the EQ-5D self-classifier can be converted into a single summary index: the EQ-5D index [1]. The EQ-5D index ranges between 0 (worst imaginable health state) and 1 (best imaginable health state). We have added this information in the Methods section of the revised manuscript (page 3, 2nd paragraph).
“The EQ-5D enables the respondent to classify his or her health according to five dimensions. These dimensions define health in terms of mobility, self-care, usual activity, pain/discomfort, and anxiety/depression. Each dimension is divided into three level, i.e., no problem/some or moderate problems/extreme problems. The information derived from the EQ-5D self-classifier can be converted into a single summary index: the EQ-5D index [1].”
Reference
- Lee, Y.K.; Nam, H.S.; Chuang, L.H.; Kim, K.Y.; Yang, H.K.; Kwon, I.S.; Kind, P.; Kweon, S.S.; Kim, Y.T. South Korean time trade-off values for EQ-5D health states: modeling with observed values for 101 health states. Value in health : the journal of the International Society for Pharmacoeconomics and Outcomes Research 2009, 12, 1187-1193, doi:10.1111/j.1524-4733.2009.00579.x.
C4. Results regressions in tables 3 and 4 are misinterpreted. for example for obstruction, people with cough were 1.69 more likely to have cough irrespective of the severity of this dysfunction. the same for the other variables introduced in your model. the same for restriction. The titles of the tables 3 for are not reflecting the content of them.
R4-1. We agree with your concern. Our original tables might be difficult to understand because they showed overall population as well as subgroups according to the severity of ventilatory disfunctions. Thus, to enhance readability, we modified Table 3 and Table 4 showing the impact of each ventilatory disorder according to the degree of its severity when compared to subjects with normal spirometry results. Additionally, we provided the analyses results of overall population separately as the Supplementary Table 1 in the revised manuscript.
R4-2. Thank you for your careful reading of our manuscript. We modified the titles of Table 3 and Table 4 to reflect the contents (pages 8–9).
C5. Table 2 does not mention for which comparison set(ie normal and obstructive? normal and restrictive? obstructive and restrictive? all three? is the p value mentioned
R5. Thank you for your comments. The p values provide in the Table 2 are comparisons of three groups (normal ventilation, obstructive ventilatory disorders, and restrictive ventilatory disorders). We have added this information in the footnotes of Tables 1–2.
Reviewer 2 Report
This study deals with a very local problem.
The information was presented in a form difficult to be followed by the reader.
The overall rewriting of the article should be provided.
Author Response
## Response to Reviewer 2’s comments
General comments
This study deals with a very local problem. The information was presented in a form difficult to be followed by the reader. The overall rewriting of the article should be provided.
Response. We understand your concern. However, tuberculosis (TB) is still one of most serious communicable disease that is a major cause of ill health and the leading cause of death from a single agent worldwide [1]; furthermore, approximately 50% of TB survivors experience post-TB lung damage and ventilatory disorders [2]. Hence, details on ventilatory disorders among subjects with prior TB histories and their impacts will be very informative to the international readers of J Pers Med when considering the scope of the journal.
References
- World Health Organization. Global tuberculosis report 2020: executive summary. 2020.
- Ravimohan, S.; Kornfeld, H.; Weissman, D.; Bisson, G.P. Tuberculosis and lung damage: from epidemiology to pathophysiology. European respiratory review : an official journal of the European Respiratory Society 2018, 27, doi:10.1183/16000617.0077-2017.
Reviewer 3 Report
I have read the manuscript by Yang et al. with great interest. The authors investigated the impact of post-TB ventilatory defects. Obstructive and restrictive ventilatory defects are associated with high symptoms burden. This known fact was tested in post-TB patients.
Comments:
- Please, specify lung function was performed pre-bronchodilator in the methods (It is only mentioned in limitations).
- Please, provide power calculations.
- I may have missed, but how do you know if the post-TB ventilatory changes were not present before TB? In such case, can we call them post-TB at all?
- Please, provide cigarette pack years for smoking history and adjust the results on this.
- Respiratory symptoms and quality of life in COPD is very much associated with medications, yet I see no data for this. Please adjust the findings on medications.
- Add data on exacerbations and hospitalisations as part of the symptoms burden.
Author Response
## Response to Reviewer 3’s comments
General comments
I have read the manuscript by Yang et al. with great interest. The authors investigated the impact of post-TB ventilatory defects. Obstructive and restrictive ventilatory defects are associated with high symptoms burden. This known fact was tested in post-TB patients.
Response. Thank you for your positive comments and a good summary. We are submitting a revised manuscript that addresses these concerns. A detailed point-by-point responses to these concerns is provided.
Specific comments
C1. Please, specify lung function was performed pre-bronchodilator in the methods (It is only mentioned in limitations).
R1. Thank you for your careful reading of our manuscript. Obstructive ventilatory disorders were defined by pre-bronchodilator spirometric results because the Korea NHANES does not provide post-bronchodilator spirometric results. We clarified this in the Methods section of the revised manuscript (page 3, 3rd paragaph).
“Since post-bronchodilator spirometry was not available in Korea NHANES database, pre-bronchodilator spirometry results were used in our study.”
C2. Please, provide power calculations.
R2. Thank you for your comments. A sample size calculation is important in terms of the statistical soundness during the design phase of the study. However, this retrospective study used the Korea NHANES database that had been established before we designed this study. To address your concern, we have acknowledged this issue as a limitation in the Discussion section of the revised manuscript (page 10, 3rd paragraph).
“Fourth, because this retrospective study used the Korea NHANES database that had been established before we designed this study, the sample size with power calculations could not be performed during the design phase of the study.”
C3. I may have missed, but how do you know if the post-TB ventilatory changes were not present before TB? In such case, can we call them post-TB at all?
R3. Thank you for pointing this out which we did not fully acknowledge in our original manuscript. As you commented, there might be some subjects who already had ventilatory disorders before M. tuberculosis infections. However, in Korea NHANES, spirometry results before TB were not available. To address your concerns, we revised our manuscript as follows: 1) we changed the title of our manuscript to “Association of Ventilatory Disorders with Respiratory Symptoms, Physical Activity, and Quality of Life in Subjects with prior Tuberculosis: A National Database Study in Korea”, 2) we changed “post-TB subjects” into “subjects with prior TB” in entire manuscript, and 3) we clarified this as a limitation in the Discussion section (pages 10, 3rd paragraph).
“Fifth, there is a possibility that some subjects already had ventilatory disorders before M. tuberculosis infection; however, in Korea NHANES database, spirometry results before TB were not available.”
C4. Please, provide cigarette pack years for smoking history and adjust the results on this.
R4. As recommended, we provided smoking amount (pack-years) for smoking history in Table 1; plus we also adjusted it for multivariable logistic and linear regression models (See Tables 1–4).
C5. Respiratory symptoms and quality of life in COPD is very much associated with medications, yet I see no data for this. Please adjust the findings on medications.
R5. Unfortunately, the Korea NHANES database does not provide data on medications. We modified the Discussion section of the revised manuscript (page 10, 3rd paragraph).
“Sixth, as the Korea NHANES database does not provide some important clinical infor-mation such as medication and exacerbation history that can affect respiratory symptoms and quality of life, we could not adjust for these factors.”
C6. Add data on exacerbations and hospitalisations as part of the symptoms burden.
R6. Because the NHANES database does not provide information on exacerbation history, we clarified this as a limitation in the Discussion section of the revised manuscript (page 11, lines 315–317). As recommended, we evaluated data on hospitalisations. However, we found a considerable number of missing values on hospitalization. Accordingly, we decided not to include data on hospitalization with discussion with our statistical team.
Round 2
Reviewer 2 Report
I am not agree with the answer, very scarsly and global. Rereading the new version I do not see the potencial of the article, even why was on the stage with Reviers. The applied All tests were two-sided, and p-values <0.05 were considered to indicate statistically significant differences. - Yes, and see this metrics only. No any other metrics has been added.
The methodology was very fuzzy and confused explaned.
Strong sttistical methods are missings, and I can't assume that the article was improved.
Generally denotes that ths article was a part of some study or something incomplete and not proper at all in the following format for the reader.
Again reject.
Author Response
## Response to Reviewer 2’s comments
General comments
Comment 1 (C1). I am not agree with the answer, very scarsly and global.
Response 1 (R1). Thank you for reviewing our manuscript. As we noted during the first revision of the manuscript, we carefully suggest that post-tuberculosis (TB) ventilatory disorders are not scarce, globally. TB is still the leading cause of death from a single epidemic disease worldwide, and TB survivors experience post-TB lung damage and ventilatory function disorders [1-2]. To address your concern, we have revised our manuscript with particular attention to these aspects.
References
- World Health Organization. Global tuberculosis report 2020: executive summary. 2020.
- Ravimohan, S.; Kornfeld, H.; Weissman, D.; Bisson, G.P. Tuberculosis and lung damage: from epidemiology to pathophysiology. European respiratory review : an official journal of the European Respiratory Society 2018, 27, doi:10.1183/16000617.0077-2017.
C2. Rereading the new version I do not see the potencial of the article, even why was on the stage with Reviers. The applied All tests were two-sided, and p-values <0.05 were considered to indicate statistically significant differences. - Yes, and see this metrics only. No any other metrics has been added. The methodology was very fuzzy and confused explaned. Strong sttistical methods are missings, and I can't assume that the article was improved. Generally denotes that ths article was a part of some study or something incomplete and not proper at all in the following format for the reader.
R2-1. Statistical analyses
Thank you for your comments. According to the reviewer’s recommendation, we provided P values comparing three groups as well as two of the three groups as well. We modified Table 1, Table 2, and the Methods of the revised manuscript (please see 2.5 Statistical analysis and the footnotes of Table 1–2).
R2-2. Extensive English revisions
A commercial English-language editing service (eWorld Editing, Eugene, OR, USA) has proofread the revised manuscript.

Reviewer 3 Report
I am happy with the changes and that the authors acknowledge limitation. I suggest acceptence.
Author Response
## Response to Reviewer 3’s comments
General comments. I am happy with the changes and that the authors acknowledge limitation. I suggest acceptence.
Response. Thank you for your feedback. We appreciate the reviewer’s helpful comments, which have substantially improved the quality of our study.
